# ENERGY-AWARE NEURAL ARCHITECTURE OPTIMIZATION WITH FAST SPLITTING STEEPEST DESCENT

## ABSTRACT

Designing energy-efficient networks is of critical importance for enabling state-of-the-art deep learning in mobile and edge settings where the computation and energy budgets are highly limited. Recently, Wu et al. (2019) framed the search of efficient neural architectures into a *continuous splitting process*: it iteratively splits existing neurons into multiple off-springs to achieve progressive loss minimization, thus finding novel architectures by gradually growing the neural network. However, this method was not specifically tailored for designing energy-efficient networks, and is computationally expensive on large-scale benchmarks. In this work, we substantially improve Wu et al. (2019) in two significant ways: 1) we incorporate the energy cost of splitting different neurons to better guide the splitting process, thereby discovering more energy-efficient network architectures; 2) we substantially speed up the splitting process of Wu et al. (2019), which requires expensive eigen-decomposition, by proposing a highly scalable Rayleigh-quotient stochastic gradient algorithm. Our fast algorithm allows us to reduce the computational cost of splitting to the same level of typical back-propagation updates and enables efficient implementation on GPU. Extensive empirical results show that our method can train highly accurate and energy-efficient networks on challenging datasets such as ImageNet, improving a variety of baselines, including the pruning-based methods and expert-designed architectures.

## 1 INTRODUCTION

Deep neural networks (DNNs) have demonstrated remarkable performance in solving various challenge problems such as image classification (e.g. Simonyan & Zisserman, 2015; He et al., 2016; Huang et al., 2017), object detection (e.g. He et al., 2017a) and language understanding (e.g. Devlin et al., 2018). Although large-scale deep networks have good empirical performance, their large sizes cause slow computation and high energy cost in the inference phase. This imposes a great challenge for improving the applicability of deep networks to more real-word domains, especially on mobile and edge devices where the computation and energy budgets are highly limited. It is of urgent demand to develop practical approaches for automatizing the design of *small, highly energy-efficient* DNN architectures that are still sufficiently accurate for real-world AI systems.

Unfortunately, neural architecture optimization has been known to be notoriously difficult. Compared with the easier task of learning the parameters of DNNs, which has been well addressed by back-propagation (Rumelhart et al., 1988); optimizing the network structures casts a much more challenging discrete optimization problem, with excessively large search spaces and high evaluation cost. Furthermore, for neural architecture optimization in energy-efficient settings, extra difficulties arise due to strict constraints on resource usage.

Recently, Wu et al. (2019) investigated similar notations of gradient descent for learning network architectures and framed the architecture optimization problem into a *continuous optimization in an infinite-dimensional configuration space*, on which novel notions of *steepest descent* can be derived for incremental update of the neural architectures. In practice, the algorithm optimizes a neural network through a cycle of *paramedic updating* and *splitting* phase. In the *parametric updating* phase, the algorithm performs standard gradient descent to reach a stable local minima; in the *splitting* phase, the algorithm expands the network by splitting a subset of exiting neurons into several off-springs in an optimal way. A key observation is that the previous local minima can be turned into

a saddle point in the new higher-dimensional space induced by splitting that can be escaped easily; thus enabling implicitly architecture space exploration and achieving monotonic loss decrease.

However, the splitting algorithm in Wu et al. (2019) treats each neuron equally, without taking into account the different amount of energy consumption incurred by different neurons, thus finding models that may not be applicable in resource-constrained environments. To close the gap between DNNs design via splitting and energy efficiency optimization, we propose an energy-aware splitting procedure that improves over Wu et al. (2019) by explicitly incorporating energy-related metrics to guild the splitting process.

Another practical issue of Wu et al. (2019) is that it requires eigen-computation of the *splitting matrices*, which causes a time complexity of $\mathcal{O}(nd^3)$ and space complexity of $\mathcal{O}(nd^2)$ when implemented exactly, where $n$ is the number of neurons in the network, and $d$ is the dimension of each neuron. This makes it difficult to implement the algorithm on GPUs for modern neural networks with thousands of neurons, mainly due to the explosion of GPU memory, thus prohibiting efficient parallel calculation on GPUs. In this work, we address this problem by proposing a fast gradient-based approximation of Wu et al. (2019), which reduces the time and space complexity to $\mathcal{O}(nd^2)$ and $\mathcal{O}(nd)$, respectively. Critically, our new fast gradient-based approximation can be efficiently implemented on GPUs, hence making it possible to split very large networks, such as these for ImageNet.

Our method achieves promising empirical results on challenging benchmarks. Compared with prior art pruning baselines that improve the efficiency by removing the least significant neurons (e.g. Liu et al., 2017; Li et al., 2017; Gordon et al., 2018), our method produces a better accuracy-flops trade-off on CIFAR-100. On the large-scale ImageNet dataset, our method finds more flops-efficient network architectures that achieve 1.0% and 0.8% improvements in top-1 accuracy compared with prior expert-designed MobileNet (Howard et al., 2017) and MobileNetV2 (Sandler et al., 2018), respectively. The gain is even more significant on the low-flops regime.

## 2 SPLITTING STEEPEST DESCENT

Our work builds upon a recently proposed *splitting steepest descent* approach (Wu et al., 2019), which transforms the co-optimization of neural architectures and parameters into a continuous optimization, solved by a generalized steepest descent on a functional space. To illustrate the key idea, assume the neural network structure is specified by a set of size parameters $\boldsymbol{m} = \{m_1, \ldots, m_K\}$, where each $m_k$ denotes the number of neurons in the $k$-th layer, or the number of a certain type of neurons. Denote by $\Theta_{\boldsymbol{m}}$ the set of possible parameters of networks of size $\boldsymbol{m}$, then $\Theta_{\boldsymbol{\infty}} = \cup_{\boldsymbol{m} \in \mathbb{N}^K} \Theta_{\boldsymbol{m}}$, which we call the configuration space, is the space of all possible neural architectures and parameters.

In this view, learning parameters of a fixed network structure is minimizing the loss inside an individual sub-region $\Theta_{\boldsymbol{m}}$. In contrast, optimizing in the overall configuration space $\Theta_{\boldsymbol{\infty}}$ admits the co-optimization of both architectures and parameters. The key observation is that the optimization in $\Theta_{\boldsymbol{\infty}}$ is in fact continuous (despite being infinite dimensional), for which (generalized) steepest descent procedures can be introduced to yield efficient and practical algorithms.

In particular, Wu et al. (2019) considered a *splitting steepest descent* on $\Theta_{\boldsymbol{\infty}}$, which consists of two phases: 1) the *parametric descent* inside each $\Theta_{\boldsymbol{m}}$ with a fixed network structure $\boldsymbol{m}$, which reduces to the typical steepest descent on parameters, and 2) the *architecture descent* crossing the boundaries of different sub-regions $\Theta_{\boldsymbol{m}}$, which, in the case of Wu et al. (2019), corresponds to "growing" the network structures by *splitting* a set of critical neurons to multiple off-springs (see Figure 1a).

From the perspective of non-convex optimization, the architecture descent across boundaries of $\Theta_{\boldsymbol{m}}$ can be viewed as escaping saddle points in the configuration space. As shown in Figure 1b, when the parametric training inside a fixed $\Theta_{\boldsymbol{m}}$ gets saturated, architecture descent allows us to escape local optima by jumping into a higher dimensional sub-region of a larger network structure. The idea is that the local optima inside $\Theta_{\boldsymbol{m}}$ can be turned into a saddle point when viewed from the higher dimensional space of larger networks (Figure 1c), which is escaped using splitting descent.

**Escaping local minima via splitting** It requires to fix a proper notion of distance on $\Theta_{\boldsymbol{\infty}}$ in order to derive a steepest descent algorithm. In Wu et al. (2019), steepest descent with $\infty$-Wasserstein

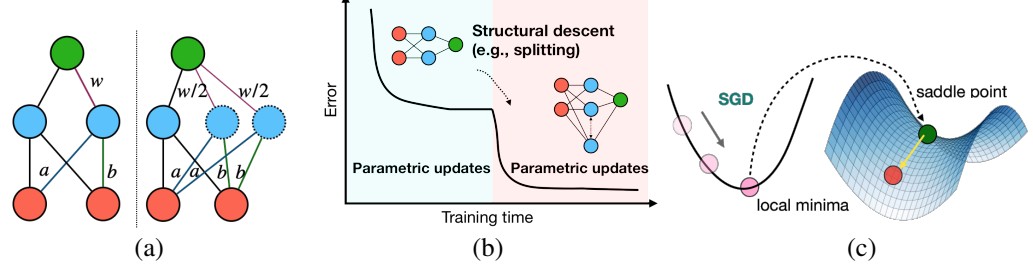

Figure 1: (a) Splitting one neuron into two off-springs. (b) Steepest descent on the overall architecture space consists of both standard gradient descent on the parameters (with fixed network structures), and updates of the network structures (via splitting). (c) The local optima in the low dimensional space is turned into a saddle point in a higher dimensional of the augmented networks, and hence can be escaped by our splitting strategy, yielding monotonic decrease of the loss.

distance was considered, which is shown to naturally correspond to the practical procedure of splitting neurons. Here we only introduce the intuitive idea from the practical perspective of optimally splitting neurons. The readers are referred to Wu et al. (2019) for more theoretical discussion.

Consider the simplest case of splitting a single neuron. Let $\sigma(\theta, x)$ be a neuron inside a neural network that we want to learn from data, where $\theta$ is the parameter of the neuron and $x$ its input variable. Assume the loss function of $\theta$ has a general form of

$$L(\theta) := \mathbb{E}_{x \sim \mathcal{D}}[\Phi(\sigma(\theta, x))], \tag{1}$$

where $\mathcal{D}$ is the data distribution, and $\Phi(\cdot)$ is the map from the output of the neuron to the final loss. In this work, the word "neuron" broadly refers to repeatable modules in neural networks, such as the typical hidden neurons, filters in CNNs.

Assume we have achieved a stable local optimum of $L(\theta)$, that is, $\nabla L(\theta) = 0$ and $\nabla_{\theta\theta} L(\theta) \succ 0$, so that we can not further decrease the loss by local descent on the parameters. In this case, splitting steepest descent enables further descent by introducing more neurons via splitting. Specifically, we split $\theta$ into $m$ off-springs $\boldsymbol{\theta} := \{\theta_i\}_{i=1}^m$, and replace the neuron $\sigma(\theta, x)$ with a weighted sum of the off-spring neurons $\sum_{i=1}^m w_i \sigma(\theta_i, x)$, where $\boldsymbol{w} := \{w_i\}_{i=1}^m$ is a set of positive weights assigned on each of the off-springs, and satisfies $\sum_{i=1}^m w_i = 1$, $w_i > 0$. See Figure 1a for an illustration. This yields an augmented loss function on $\boldsymbol{\theta}$ and $\boldsymbol{w}$:

$$\mathcal{L}_m(\boldsymbol{\theta}, \boldsymbol{w}) := \mathbb{E}_{x \sim \mathcal{D}}\left[\Phi\left(\sum_{i=1}^m w_i \sigma(\theta_i, x)\right)\right]. \tag{2}$$

It is easy to see that if we set $\theta_i = \theta$ for all the off-springs, the network remains unchanged. Therefore, as we change $\theta_i$ in a small neighborhood of $\theta$, it introduces a smooth change on the loss function. Splitting steepest descent is derived by considering the optimal splitting strategies to achieve the steepest descent on loss in a small neighborhood of the original parameters.

**Deriving splitting steepest descent** Derive the optimal splitting strategy involves deciding the number of off-springs $m$, the values of the weights $\{w_i\}$ and the parameters for the off-springs $\{\theta_i\}$. In Wu et al. (2019), this is formulated into the following optimization problem:

$$\min_{m, \boldsymbol{\theta}, \boldsymbol{w}} \left\{ \mathcal{L}_m(\boldsymbol{\theta}, \boldsymbol{w}) - L(\theta) \quad \text{s.t.} \quad ||\theta_i - \theta|| \le \epsilon, \ \sum_{i=1}^m w_i = 1, \ w_i > 0, \ \forall i \in [m], \ m \in \mathbb{N}_+ \right\}, \tag{3}$$

where the parameters $\boldsymbol{\theta} := \{\theta_i\}_{i=1}^m$ of the off-springs are restricted within an infinitesimal $\epsilon$-ball of the original parameter $\theta$, that is, $||\theta_i - \theta|| \le \epsilon$, with $\epsilon$ a small positive step size parameter. Note that the number of off-springs $m$ is also optimized, yielding an infinite dimensional optimization.

Fortunately, when $\epsilon$ is very small, the optimum of Equation (3) is achieved by either $m = 1$ (no splitting) or $m = 2$ (two off-springs). The property of the optimal solution is characterized (asymptotically) by the following key *splitting matrix* $S(\theta)$,

$$S(\theta) = \mathbb{E}_{x \sim \mathcal{D}}\left[\nabla_\sigma \Phi(\sigma(\theta, x)) \nabla_{\theta\theta}^2 \sigma(\theta, x)\right],$$

which is a symmetric $d \times d$ matrix (and $d$ is the dimension of $\theta$). The optimum of Equation (3), when $L(\theta)$ reaches a stable local optimum (i.e., $\nabla_\theta L(\theta) = 0$, $\nabla_{\theta\theta} L(\theta) \succ 0$), is determined by $S(\theta)$ via

$$\min_{m, \boldsymbol{\theta}, \boldsymbol{w}} \left\{ \mathcal{L}_m(\boldsymbol{\theta}, \boldsymbol{w}) - L(\theta) \right\} = \frac{\epsilon^2}{2} \min \left\{ \lambda_{\min}(S(\theta)), \, 0 \right\} \, + \, \mathcal{O}(\epsilon^3), \tag{4}$$

where $\lambda_{\min}(S(\theta))$ denotes the minimum eigenvalue of $S(\theta)$, and it is called the *splitting index*.

When $\lambda_{\min}(S(\theta)) > 0$, the loss can not be improved by any splitting strategies following (4). When $\lambda_{\min}(S(\theta)) < 0$, the maximum decrease of loss, which equals $\epsilon^2 \lambda_{\min}(S(\theta))/2$, can be achieved by a simple strategy of *splitting the neuron into two copies with equal weights*, whose parameters are updated along the minimum eigen-vectors $v_{\min}(S(\theta))$ of $S(\theta)$, that is,

$$m = 2, \qquad \theta_1 = \theta + \epsilon v_{\min}(S(\theta)), \qquad \theta_2 = \theta - \epsilon v_{\min}(S(\theta)), \qquad w_1 = w_2 = 1/2. \tag{5}$$

In this case, splitting allows us to escape the parametric local optima to enable further improvement.

**Splitting deep neural networks**  As shown in Wu et al. (2019), the result above can be naturally extended to more general cases when we need to split multiple neurons in deep neural networks. Consider a neural network with $n$ neurons $\theta^{[1:n]} = \{\theta^1, \cdots, \theta^n\}$. Assume we split a subset $A$ of neurons with the optimal strategy in Equation (5) following their own splitting matrices, the improvement of the overall loss equals the sum of individual gains:

$$G(A) = \sum_{\ell \in A} \lambda_{\min}(\ell),$$

where $\lambda_{\min}(\ell) := \lambda_{\min}(S_\ell(\theta^\ell))$ denotes the minimum eigenvalue of the splitting matrix $S_\ell(\theta^\ell)$ associated with neuron $\ell$. Therefore, given a budget of splitting at most a given number of neurons, the optimal subset of neurons to split are the top ranked neurons with the smallest, and negative minimum eigenvalues. Overall, the splitting descent in Wu et al. (2019) alternates between parametric updates with fixed network architectures, and splitting top ranked neurons to augment the architectures, until a stopping criterion is reached.

## 3  NEURAL ARCHITECTURE OPTIMIZATION VIA ENERGY-AWARE SPLITTING

The method above allows us to select the best subset of neurons to split to yield the steepest descent on the loss function. In practice, however, splitting different neurons incurs a different amount of increase on the model size, computational cost, and physical energy consumption. For example, splitting a neuron connecting to a large number of inputs and outputs increases the size and computational cost of the network much more significantly than splitting the neurons with fewer inputs and outputs. In practice, convolutional layers close to inputs often have larger feature maps which lead to a high energy cost, and layers closer to outputs have smaller feature maps and hence lower computational cost. A better splitting strategy should take the cost of different neurons into account.

To address this problem, we propose to explicitly incorporate the energy cost to better guide the splitting process. Specifically, for a neural network with $n$ neurons, we propose to decide the optimal splitting set by solving the following constrained optimization:

$$\min_{\boldsymbol{\beta}} \sum_{\ell=1}^{n} \beta_\ell \lambda_{\min}(\ell), \quad \text{s.t.} \;\; \sum_{\ell=1}^{n} e_\ell \beta_\ell \leq e_{budget}, \;\; \beta_\ell \in \{0, 1\}, \;\; \forall \ell \tag{6}$$

Here $\boldsymbol{\beta} \in \mathbb{R}^n$ is a binary mask, with $\beta_\ell$ indicates whether the $\ell$-th neuron should be split ($\beta_\ell = 1$) or not ($\beta_\ell = 0$), and $e_\ell$ represents the cost of splitting at the current iteration. We search for the optimal subset of neurons that yields the largest descent on the loss (in terms of the splitting index), while incurring a total energy cost no larger than a budget threshold $e_{budget}$. This optimization Equation (6) is a standard *knapsack problem*. The exact solution of knapsack problems can be very expensive due to their NP-hardness. In practice, we use linear programming relaxation for fast approximation by relaxing the integrality constrains to linear constrains such that $\beta_\ell \in [0, 1], \forall \ell$. The continuous relaxation could then be solved using standard linear programming tools efficiently (Dantzig, 1998). Finally, we define the optimal splitting set $A := \{\beta_\ell > 0.9, \forall \ell\}$. For each neuron in $A$, we split it into two equally weighted off-springs along their splitting gradients, following Equation (5).

In this work, we take $e_\ell$ to be the energy cost, and estimate it by the increase of flops if we split the $\ell$-th neuron starting from the current network structure. Note that the cost of splitting the same neuron changes when the network size changes across iterations. Therefore, we re-evaluate the cost of every neuron at each splitting stage, based on the architecture of the current network.

## 4 FAST SPLITTING WITH RAYLEIGH-QUOTIENT GRADIENT DESCENT

A practical issue of splitting steepest descent is the high computational cost of the eigen-computation of the splitting matrices. The time complexity of evaluating all splitting indexes is $\mathcal{O}(nd^3)$. Here $n$ is the number of neurons and $d$ is the number of the parameters of each neuron. Meanwhile, the space complexity is $\mathcal{O}(nd^2)$. Although this is manageable for networks with small or moderate sizes, an immediate difficulty for modern deep networks with thousands of high-dimensional neurons ($\sim 1000$) is that we are not able to store all splitting-matrices on GPUs, which necessities slow calculation on CPUs. It is desirable to further speed up the calculation for very large scale problems. In this section, we propose an approach for computing the splitting indexes and gradients *without explicitly expanding the splitting matrices*, based on fast (stochastic) gradient descent on the Rayleigh quotient.

**Rayleigh-Quotient Gradient Descent for Minimum Eigenvalues**   The key idea is to note that the minimum eigenvalues and eigenvectors of a matrix $S \in \mathbb{R}^{d \times d}$ can be obtained by minimizing the Rayleigh quotient (Parlett, 1998),

$$\lambda_{min} = \min_v \left\{ \mathcal{R}_S(v) := \frac{v^\top S v}{v^\top v} \right\}, \qquad v_{min} \propto \arg\min_v \mathcal{R}_S(v), \qquad (7)$$

which can be solved using gradient descent or other numerical methods. Although this problem is non-convex, $v_{min}$ can be shown to be the unique global minimum of $R(v)$, and all the other stationary points, corresponding to the other eigenvectors, are saddle points and can be escaped with random perturbation. Therefore, stochastic or noisy gradient descent on $R(v)$ is expected to converge to $v_{min}$. The gradient of $R(v)$ w.r.t. $v$ can be written as follows,

$$\nabla_v \mathcal{R}_S(v) = 2 \|v\|^{-2} \left( Sv - \mathcal{R}_S(v)v \right) \propto Sv - \mathcal{R}_S(v)v,$$

which depends on $S$ only through the matrix-vector product $Sv$. A significant saving in computation can be obtained by directly calculating $Sv$ at each iteration, without explicitly expanding the whole matrix. This can be achieved by the following auto-differentiation trick.

**Automatic Differentiation Trick**   Recall that the splitting matrix of a single neuron is $S(\theta) = \mathbb{E}_{x \sim \mathcal{D}}[\nabla_\sigma \Phi(\sigma(\theta, x)) \nabla^2_{\theta\theta} \sigma(\theta, x)]$. To calculate $S(\theta)v$ for any vector $v \in \mathbb{R}^d$, we construct the following auxiliary function,

$$F(\eta) = \mathbb{E}_{x \sim \mathcal{D}}[\Phi(\sigma(\theta, x) + \eta^\top \nabla^2_{\theta\theta} \sigma(\theta, x)v))], \qquad \eta \in \mathbb{R}^d,$$

with which it is easy to show that $S(\theta)v = \nabla_\eta F(0)$. Here $F(\eta)$ corresponds to simply adding an extra term on the top of the neuron's output and can be constructed conveniently.

In the case of deep neural networks with $n$ neurons $\{\theta^\ell\}_{\ell=1}^n$, we can calculate all the matrix-vector product $\{g_\ell := S_\ell(\theta^\ell)v_\ell\}_{\ell=1}^n$ for all the neurons jointly with a single differentiation process. More precisely, for each neuron $\theta^\ell$, we can add a term $\eta_\ell^\top \nabla^2_{\theta\theta} \sigma_\ell(\theta^\ell, x)v_\ell$ (denoted as *auxiliary activation*) on its own output (see Figure 2). Thus, we obtain a joint function $F(\eta_1, \ldots, \eta_n)$, for which it is easy to see that $\nabla_{\eta_\ell} F(\eta_1, \ldots, \eta_n) = g_\ell$, $\forall \ell$. Therefore, simply differentiating $F(\eta_1, \ldots, \eta_n)$ allows us to obtain all $\{g_\ell\}$ simultaneously.

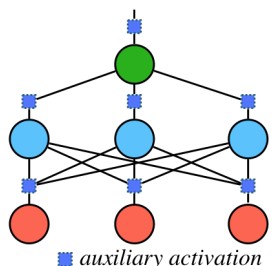

*auxiliary activation*

Figure 2: Illustration of the auto-differentiation trick with auxiliary activation.

**Stochastic Gradient on Rayleigh quotient**   Note that we still need to average over the whole dataset $\mathcal{D}$ to measure the Rayleigh quotient gradients $\{g_\ell\}$, this is computationally expensive in the

---

**Algorithm 1** Energy-aware neural architecture optimization with fast splitting steepest descent

---

Starting from a small base network (viewed as the "seed"), we gradually grow the neural network by alternating between the following two phases until an energy constrain reached:

**1. Parametric Updates**: Optimize neuron weights using standard methods (e.g., SGD) until no further improvement can be made by only updating parameters.

**2. Splitting Neurons**:
   (a) Computing the splitting index of each neuron using gradient-based approximation (Section 4);
   (b) Finding the optimal set of neurons to split by solving the energy-aware allocation problem in Equation (6);
   (c) For each neuron selected above (step 2a), split it into two equally weighted off-springs along their splitting gradients, following Equation (5).

---

case of big data. However, we can conveniently address this by approximating $\{g_\ell\}$ with subsampled mini-batches $\mathcal{B}$. In the case of single-neuron networks, that is,

$$Sv = \nabla_\eta \hat{F}(\eta), \ \ \text{with} \ \ \hat{F}(\eta) = \frac{1}{|\mathcal{B}|} \sum_{i=1}^{|\mathcal{B}|} \left[ \Phi\left( \sigma(\theta, x_i) + \eta^\top \nabla^2_{\theta\theta} \sigma(\theta, x_i) v \right) \right].$$

Assume we sweep the training data $T$ times to train the Rayleigh-Quotient to convergence (see Equation (7)). In this way, the splitting time complexity for approximating all splitting indexes and gradients would be only $\mathcal{O}(Tnd^2)$ ($T$ is often a small constant). More importantly, a significant advantage of our gradient-based approximation is that the space complexity is only $O(nd)$. In this way, all calculation could be efficient performed on GPUs. This given us an algorithm for splitting that is almost as efficient as back-propagation.

**Overall Algorithm** Our overall algorithm in shown in Algorithm 1, which improves over Wu et al. (2019) by offering much lower time and space complexity, and the flexibility of incorporating energy and other costs of different neurons. It can be implemented easily using modern deep learning frameworks such as Pytorch (Paszke et al., 2017). Our code is available with the submission.

## 5 EXPERIMENTS

We apply our method to split small variants of MobileNetV1 (Howard et al., 2017) and MobileNetV2 (Sandler et al., 2018), on both CIFAR-100[1] and ImageNet dataset. We show our method finds networks that are more accurate and also more energy-efficient compared to expert-designed architectures and pruned models.

**Settings of Our Algorithm** In all our tests of our Algorithm 1, we restrict the increase of the energy cost to be smaller than a budget $e_{budget}$ at each splitting stage. We set $e_{budget}$ adaptively to be proportional to the total flops of the current network such that the flops of the augmented network obtained by splitting cannot exceed $1 + \alpha$ times of the previous one. We denote by $\alpha$ the growth ratio and set $\alpha = 0.5$ unless otherwise specified.

For our fast *splitting indexes* approximation (see section 4), we set batch size $|\mathcal{B}|$ to be 64 and use RMSprop (Tieleman & Hinton, 2012) optimizer with 0.001 learning rate. We find the Rayleigh-Quotient converges fast in general: for small CIFAR-10/100 datasets, we train 10 epochs (T=10); for the large-scale ImageNet set, we find a small T (=2) is sufficient.

### 5.1 TESTING IMPORTANCE OF ENERGY-AWARE SPLITTING (RESULTS ON CIFAR-10)

To study the importance of our energy-aware splitting, we compare our method (denoted as *splitting (energy-aware)*) to Wu et al. (2019) (denoted as *splitting (vanilla)*), which doesn't use energy metrics to guide the splitting process. In this experiment, we apply both splitting algorithms to grow a variant

---

[1]https://www.cs.toronto.edu/~kriz/cifar.html

of small version of MobileNets (Howard et al., 2017) trained on the CIFAR-10 dataset, in order to test the importance of using energy cost for splitting.

**Settings**  We test our algorithm on two variants of MobileNet, each of which consists one regular $3 \times 3$ convolution layer, followed by $k = 3$ and $k = 6$ MobileNet blocks (Howard et al., 2017), respectively. In both variants, the resolutions are reduced 3 times evenly and one extra MobileNet block attached with a fully connected layer for classification. Note that each MobileNet block consists a depthwise convolutional layer and a pointwise convolutional layer. In our implementation, we only split the convolutional filters in the pointwise convolutional layers and duplicate the corresponding depthwise convolution filters accordingly during splitting. We start with small networks that have the same number of channels (=8) across all layers to better study the behavior of how neurons are split. We set batch size to be 256 and learning rate 0.1 for 160 epochs, with learning rate dropped 10x at 80 and 120 epochs for the two variants ($k = 3, 6$), respectively.

**Results**  Our results are shown in Figure 3, which shows that our *splitting (energy-aware)* approach yields better trade-offs of accuracy and flops than *splitting (vanilla)* in both cases ($k = 3$ and $k = 6$). We find that *splitting (vanilla)* does discover networks with small model size (fewer parameters, see Figure 3 (b) and (d)), but does not yield lower energy consumption in practice. These results highlight the importance of using real energy cost for guiding the splitting process in order to optimize for the best energy-efficiency.

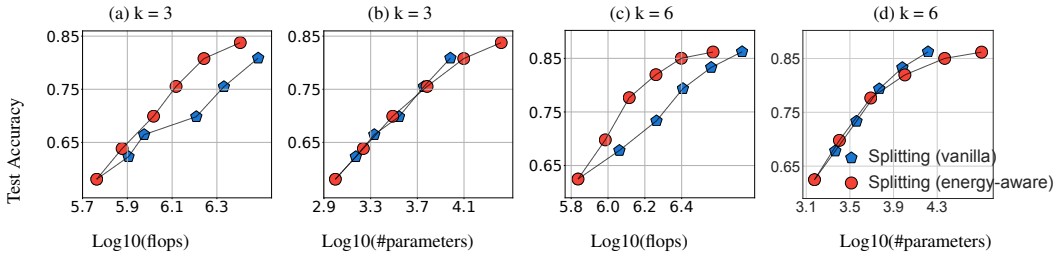

Figure 3: Comparisons between our energy-aware splitting and standard splitting in Wu et al. (2019) on CIFAR-10. Results are shown for two variants of MobileNet, one with $k = 3$ (4 MobileNet blocks, 9 layers in total), another with $k = 6$ (7 MobileNet blocks, 15 layers in total).

## 5.2  RESULTS ON CIFAR-100

We compare our method with several state-of-the-art pruning baselines on the CIFAR-100 dataset. We also show our fast gradient-based splitting approximation in section 4 achieves the same accuracy as the exactly eigen-computation, while significantly reducing the overall splitting time.

**Settings**  We again apply splitting on a small version of MobileNet (Howard et al., 2017) (with the same network topology) to obtain a sequence of increasingly large models. Specifically, we set the number of channels of the base model to be $2, 4, 8, 8, 16, 16, 24, 24, 24, 24, 24, 24, 32, 32$ for each layer, respectively. We compare our method with a simple but competitive *width multiplier* (Howard et al., 2017) baseline, which prunes filters uniformly across layers (denoted as *Width multiplier*) from the original full size MobileNet. We also experiment with three state-of-the-art structured pruning methods: *Pruning (Bn)* (Liu et al., 2017), *Pruning (L1)* (Li et al., 2017) and *MorphNet* (Gordon et al., 2018). The implementation of all the baselines are based on Liu et al. (2019b). For all methods, we normalize the inputs using channel means and standard deviations. We use stochastic gradient descent with momentum 0.9, weight decay 1e-4, batch size 128. We set 0.1 initial learning rate for 160 epochs, with learning rate decreased by 10x at epochs 80, 120, respectively. For all pruned models, we report the finetune performance with the same training settings. For Morphnet, we grid search the best sparsity hyper-parameter $\lambda$ in the range [1e-8, 5e-8, 1e-9, 5e-9, 1e-10] and report the best models found.

**Results**  Figure 4 (a) shows the results on CIFAR-100, in which our method achieves the best accuracy when targeting similar flops. To draw further comparison between the splitting and pruning

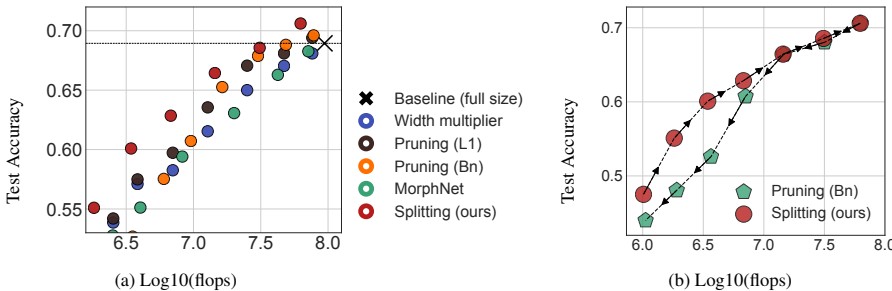

Figure 4: (a) Results on CIFAR-100 using MobileNet(Howard et al., 2017); (b) we show our energy-aware splitting approach can learn more accurate and energy-efficient (with small flops) networks than pruning methods (Liu et al., 2017).

approaches, we prune the final network learned by our splitting algorithm to obtain a sequence of increasingly smaller models using *Pruning (Bn)* (Liu et al., 2017). As shown in Figure 4 (b), it is clear that our splitting checkpoints (red circles) form a better flops-accuracy trade-off curve than models obtained by pruned from the same model (green Pentagons). This confirms the advantage of our method in neural architecture optimization, especially on the low-flops regime.

In Figure 5 (a-b), we examine the accuracy and speed of our fast gradient-based eigen-approximation. We run all methods on a server with one V100 GPU and 16 CPU cores and report the wall-clock time. We can see that our fast method (red dots and bars) achieves almost the same accuracy as the splitting based on exact eigen-decomposition (blue dots and bars), while achieving significant gain in computational time (see Figure 5 (b)).

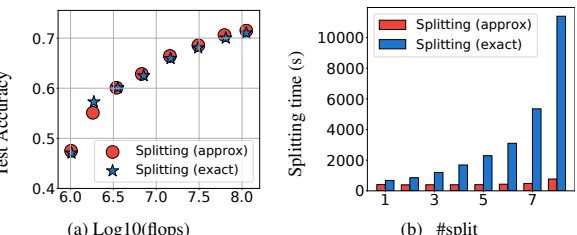

Figure 5: Comparison of testing accuracy (a) and splitting time (b) using exact eigen-decomposition (denoted as *splitting (exact)*) and our fast gradient-based eigen-approximation (denoted as *splitting (approx)*).

### 5.3 RESULTS ON IMAGENET

We conduct experiments on large-scale ImageNet dataset, on which our method again shows clear advantages over existing methods. Note that splitting based on exact eigen-composition is no longer feasible on ImageNet and our fast gradient-based approximation must be used.

**Dataset** The ImageNet dataset (Deng et al., 2009) consists of about 1.2 million training images, and $50,000$ validation images, classified into $1,000$ distinct classes. We resize the image size to $224 \times 224$, and adopt the standard data augment scheme (mirroring and shifting) for training images (e.g. Howard et al., 2017; Sandler et al., 2018).

**Settings** We choose both MobileNetV1 (Howard et al., 2017) and MobileNetV2 (Sandler et al., 2018) as our base net for splitting, which are strong baselines and specifically hand-designed and heavily tuned to optimize accuracy under a flops-constrain on the ImageNet dataset.

For parametric updates, we follow standard training settings on the ImageNet dataset using MobileNets. Specifically, we train with a batch-size of $128$ on 4 GPUs (total batch size $512$). We use stochastic gradient descent with an initial learning rate $0.2$ and $0.1$ for MobileNetV1 and MobileNetV2, respectively. We apply cosine learning rate annealing scheduling and use label smoothing ($0.1$) by following (He et al., 2019).

For our method, we start with relative small models (denoted by *Splitting-0 (seed)*) by shrinking the network uniformly with a width multipler $0.3$, and gradually grow the network via energy-aware splitting. We use Splitting-$k$ to represent the model we discovered at the $k$-th splitting stage. We report the single-center-crop validation error of different models.

| Model | MACs (G) | Top-1 Accuracy | Top-5 Accuracy |
|---|---|---|---|
| MobileNetV1 (1.0x) | 0.569 | 72.93 | 91.14 |
| Splitting-4 | 0.561 | **73.96** | **91.49** |
| MobileNetV1 (0.75x) | 0.317 | 70.25 | 89.49 |
| AMC (He et al., 2018) | 0.301 | 70.50 | 89.30 |
| Splitting-3 | **0.292** | **71.47** | **89.67** |
| MobileNetV1 (0.5x) | 0.150 | 65.20 | 86.34 |
| Splitting-2 | **0.140** | **68.26** | **87.93** |
| Splitting-1 | 0.082 | 64.06 | 85.30 |
| Splitting-0 (seed) | 0.059 | 59.20 | 81.82 |

Table 1: Results of ImageNet classification using MobileNetV1. Splitting-$k$ denotes the model we discovered at the $k$-th splitting stage. Our method yields networks with both higher accuracy and lower number of multiply-and-accumulate (MAC) operations.

| Model | MACs (G) | Top-1 Accuracy | Top-5 Accuracy |
|---|---|---|---|
| MobileNetV2 (1.0x) | 0.300 | 72.04 | 90.57 |
| Splitting-3 | **0.298** | **72.84** | **90.83** |
| MobileNetV2 (0.75x) | 0.209 | 69.80 | 89.60 |
| AMC (He et al., 2018) | 0.210 | 70.85 | 89.91 |
| Splitting-2 | **0.208** | **71.76** | **90.07** |
| MobileNetV2 (0.5x) | 0.097 | 65.40 | 86.40 |
| Splitting-1 | **0.095** | **66.53** | **87.00** |
| Splitting-0 (seed) | 0.039 | 55.61 | 79.55 |

Table 2: Results on ImageNet using MobileNetV2. Splitting-$k$ denotes the model we discovered at the $k$-th splitting stage. MAC denotes the number of multiply-and-accumulate operations.

**MobileNetV1 Results**   In Table 1, we find that our method achieves about $1\%$ top-1 accuracy improvements in general when targeting similar flops. On low-flops regime ($< 0.15$G flops), our method achieves 3.06% higher top-1 accuracy compared with MobileNet (0.5X) (with width multiper 0.5). Also, the model found by our method is $0.97\%$ higher than a prior art pruning method (*AMC* (He et al., 2018)) when comparing with checkpoints with $\sim 0.3$G flops.

**MobileNetV2 Results**   From table 2, we find that our splitting models yield better performance compared with prior art expert-designed architectures on all flops-regimes. Specially, out *splitting-3* reaches 72.84 top-1 accuracy; this yields an 0.8% improvement over its corresponding baseline model. On the low-flops regime, our *splitting-2* achieves an 1.96% top-1 accuracy improvement over MobileNetV2 (0.75x); our *splitting-1* is 1.1% higher than MobileNetV2 (0.5x). Our performance is also about 0.9% higher than AMC when targeting 70% flops.

## 5.4   ABLATION STUDY

In our algorithm, the growth ratio $\alpha$ controls how many neurons we could split at each splitting stage. In this section, we perform an in-depth analysis of the effect of different $\alpha$ values. We also examine the robustness of our splitting method regarding randomness during the network training and splitting (e.g. parameters initializations, data shuffle).

**Impact of growth ratio**   To find the optimal growth ratio $\alpha$, we ran multiple experiments with different growth ratio $\alpha$ under the same settings as section 5.2. Figure 6 (a) shows the performance of various runs. We find that the growth ratio in the range of $[0.3, 0.5]$ tend to perform similarly well. However, the smaller growth ratio of $\alpha = 0.2$ tends to give lower accuracy, this may be because with a small growth ratio, the neurons in the layers close to the input may never be selected because of their higher energy cost for splitting, hence yielding sub-optimal networks.

**Robustness** We apply our method to grow a small MobileNet (Howard et al., 2017) using different random seeds for parameters initialization and data shuffle under the same setting as Figure 6 (a) with a growth ratio 0.5. Figure 6 (b) shows the test performance of different models learned. As we can see from Figure 6 (b), all runs perform similarly well with small variations.

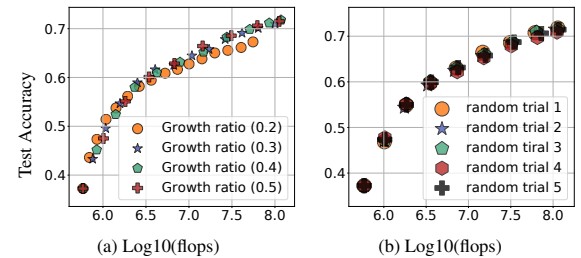

Figure 6: Comparison on the test accuracy for various ablation study settings.

## 6 RELATED WORK

Neural architecture search (NAS) has been found a powerful tool for automating energy-efficient architecture design. Most existing NAS methods are based on black-box optimization techniques, including reinforcement learning (e.g. Zoph & Le, 2017; Zoph et al., 2018) , evolutionary algorithms (e.g. Real et al., 2019; 2017). However, these methods are often extremely time-consuming due to the enormous search space of possible architectures and the high cost for evaluating the performance of each candidate network. More recent approaches have made the search more efficient by using weight-sharing (e.g. Pham et al., 2018; Liu et al., 2019a; Cai et al., 2019), which, however, suffers from the so-called multi-model forgetting problem (Benyahia et al., 2019) that causes training instability and performance degradation during search. Overall, designing the best architectures using NAS still requires a lot of expert knowledge and trial-and-errors.

In contrast, pruning-based methods construct smaller networks from a pretrained over-parameterized neural network by gradually removing the least important neurons. Various pruning strategies have been developed based on different heuristics (e.g., Han et al., 2016; Li et al., 2017; Luo et al., 2017; He et al., 2017b; Peng et al., 2019), including energy-aware pruning methods that use energy consumption related metrics to guide the pruning process (e.g., Yang et al., 2017; Gordon et al., 2018; He et al., 2018; Yang et al., 2019). However, a common issue of these methods is to alter the standard training objective with sparsity-induced regularization which necessities sensitive hyperparameters tuning. Furthermore, the final performance is largely limited by the initial hand-crafted network, which may not be optimal in the first place.

## 7 CONCLUSIONS

In this work, we present a fast energy-aware splitting steepest descent approach for resource-efficient neural architecture optimization that generalizes Wu et al. (2019). Empirical results on large-scale ImageNet benchmark using MobileNetV1 and MoibileNetV2 demonstrate the effectiveness of our method.

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
