# OpenReview forum: "Energy-Aware Neural Architecture Optimization with Fast Splitting Steepest Descent"
_ICLR.cc/2020/Conference — Reject_

### Official Review · AnonReviewer1 · 2019-10-24
**Official Blind Review #1**

**Rating:** 3

**Review:**

This paper is based on a prior work which proposed Splitting Steepest Descent to search better network structures via splitting existing neurons to multiple off-springs. As an improvement, the authors (1) incorporate the energy cost to better guide the splitting process and (2) reduce the time and space complexity by approximating the original computation process with Rayleigh-Quotient Gradient Descent. They conduct experiments on public image classification datasets using lightweight networks as backbones to show that their algorithm outperforms existing methods. The paper is well written and easy to follow. The experiments are comprehensive and the evaluation results shows good properties of proposed method.

In brief, this paper is an improvement to a splitting algorithm in a previous work, achieving good efficiency and enabling application on large datasets. However, the theory needs more justification and the experiments results are not sufficient to show the significance of their contribution. Therefore, this paper may not be accepted unless more experiment results are given.

For the theory & modeling, the following should be addressed.
1.	The mathematical justification of the optimization on energy cost is not very sound, and the definition of optimal splitting set seems arbitrary.
2.	Given that the experiments are conducted on convolutional networks, it would be more illustrative if the paper describe the process of applying the algorithm on common convolutional operators.
3.	The novelty is limited by the prior work.

For the experiment, the following should be addressed.
1.	The experiments are mainly conducted on MobileNet network. It would be more convincing if more experiment is done on other lightweight or normal convolutional networks.
2.	This paper reduces time and space complexity of the algorithm in a previous work, but there is no running time or memory footprint statistics to support this argument.
3.	The paper only lists one pruning-based method as comparison in the experiments. It would be more convincing if more pruning and splitting methods are presented.


**Experience Assessment:**

I do not know much about this area.

**Review Assessment: Checking Correctness Of Derivations And Theory:**

I assessed the sensibility of the derivations and theory.

**Review Assessment: Checking Correctness Of Experiments:**

I assessed the sensibility of the experiments.

**Review Assessment: Thoroughness In Paper Reading:**

I read the paper at least twice and used my best judgement in assessing the paper.

---

> ### Author Response · Authors · 2019-11-15
> **Response to Review #1 (1)**
>
> We thank R#1 for the time and comments. Here are our responses to your comments. We hope the reviewer can kindly reconsider your evaluation based on the potential value that this work may bring to the area of energy-aware architecture optimization.
>
> Here are our replies to your problems. We hope our work can win stronger support from you to allow us to contribute better to the field. Please let us know if anything can be done to help clarify.
>
> Question: "The mathematical justification of the optimization on energy cost is not very sound, and the definition of the optimal splitting set seems arbitrary."
>
> Reply: The definition splitting set is not arbitrary. It is the most natural and simple formulation we can derive: If we formulate the problem of minimizing the loss function subject to a fixed increase of budget, our formulation we can derive by using the splitting Taylor expansion by the original splitting descent paper. The authors have investigated the problem throughout and could not another formulation that is better and simpler than the current one.
>
> We would remark that energy-aware architecture optimization is quite complicated the empirical problem, which makes most existing works rather heuristic. We believe our work based on splitting steepest descent is already "quite theoretically motivated" compared to many other works.
>
> Question: "novelty by prior work"
>
> [Since the other reviewers have the same question, we use the same answer for all the reviewers here]
>
> We understand the perception that our work does not feel novel compared to w.r.t. the original splitting descent paper. But splitting descent is a very new and recent work, with the key weakness on scalability to very large models. Our work is a timely contribution to this urgent issue. Also, our algorithm is indeed a very novel approach for the problem of energy-aware architecture optimization (no algorithms of a similar kind exist for this problem; original splitting descent was not energy-aware).  We also believe it is non-trivial to put together our techniques to nicely solve the scalability problem, including our knapsack formulation, Rayleigh gradient, and the ideas of backpropagating on the auxiliary network that allows us to avoid using for loops. We think the application of these techniques in the new content is novel.
>
> More elaboration:
>
> 1) Compared with the original splitting descent paper, our work is *very important*. The original splitting descent *does not work* for large scale settings like ImageNet due to the computation and memory constraint. Our work addresses this key issue nicely and timely. Without our work, splitting descent (and all the potential new algorithms that it may imply) can not work in very large scale settings. We hope the reviewer can kindly consider the fact that methodological novelty is not the only factor to judge the value of a paper (especially when it comes to deep learning). If every work like ours gets rejected, the field can not grow.
>
> 2) Our work is *very novel* to the field of energy-aware architecture optimization, which uses energy metrics to improve DNNs structures. All most existing works are either black-box algorithms (e.g., evolutionary, or reinforcement learning) or using network pruning. Methods that split and grow neurons progressively like our method is *very un-common* in the field of energy-aware deep learning. And we are *the first work* that shows spitting/growing-based methods indeed work empirically for energy-aware NAS on large scale.  We truly want to share this new message with the community and hope it motivates new works and ideas. We would appreciate the reviewers can kindly take this into consideration.
>
> 3) Our key techniques, including the knapsack formulation, the Rayleigh-quotient gradient, and the auto-differentiation trick, are all essential steps for making splitting descent scalable. Yes, these are existing techniques from textbook, but putting them together to solve this problem is new (since there have been no similar works like ours in the literature). We do not believe it is obviously trivial to derive and put all of them in the right place. For example, consider our step of constructing the auxiliary network using auto-diff to calculate the Rayleigh gradient all the neurons simultaneously without using for loop. We think it is indeed a new and essential step that allows us to achieve high scalability (which is at least not obvious to the authors themselves in the beginning).

---

> ### Author Response · Authors · 2019-11-15
> **Response to Review #1 (2)**
>
> *** Experiments ***
>
> Question: "The experiments are mainly conducted on MobileNet network. It would be more convincing if more experiment is done on other lightweight or normal convolutional networks. Why do we choose MobileNet as our testbed?"
>
> The goal of our algorithm is to learn small networks that are fast and energy-efficient in the inference phase. Traditional CNNs such as VGG and ResNets are often much more computationally expensive than MobileNets, which are not suitable for resource-constrained mobile and edge settings. For example, the flops of VGG-16 is about 25x larger than MobileNetV1 on the ImageNet dataset. In this paper, we focus on MobileNets due to their nice accuracy and energy trade-off.
>
> Question: "This paper reduces time and space complexity of the algorithm in previous work, but there is no running time or memory footprint statistics to support this argument"
>
> Our Rayleigh quotient based approximation is fast, 1) our method reduces the time complexity of the splitting process from O(nd^3) to O(nd^2). From Figure 5b, we can see that our approach is about 20X faster compared to exact eigen-computation when the network is large. The results clearly demonstrate the efficiency of our approach; 2) our method reduces the memory footprint from O(nd^2) to O(nd). Here d is the dimension of each neuron. Theoretically, our method achieves d times (d is normally greater than 100 for modern neural networks) memory reduction, which allows us to implement the algorithm on GPUs. We refer R#2 to the introduction and section 4 for more detailed discussions.
>
>
> Question: "The paper only lists one pruning-based method as a comparison in the experiments. It would be more convincing if more pruning and splitting methods are presented"
>
> On the CIFAR100 benchmark, we compared with several strong pruning baselines, including 1) a weight-magnitude based pruning approach (Pruning (L1)); 2) a batch-normalization based pruning approach (Pruning (Bn)); 3) an energy-aware pruning approach (MorphNet); 4) and the width-multiplier baseline. Our method outperforms all baselines. The results clearly demonstrate the effectiveness of our approach.
>
> On the large scale ImageNet dataset, we note that the original MobileNets are strong baselines, which are specifically heavily tuned to yield a good accuracy and flops trade-off. And to the best of our knowledge, AMC is the current SOTA pruning approach for MobileNets compression.

---

### Official Review · AnonReviewer2 · 2019-10-28
**Official Blind Review #2**

**Rating:** 6

**Review:**

Summary:
This paper builds on a recently proposed algorithm ("splitting steepest descent", Wu et al 2019) for guiding the growth of a smaller network into a larger one in architecture search. The algorithm in Wu, et al. alternates between two steps, (i) optimization of parameters for a fixed model and (ii) modification of the architecture by identifying a subset of neurons to split into more neurons, based on the "splitting index" of each neuron (amounting to evaluating the smallest eigenvalue of a matrix). This work builds on that in two ways: (i) it incorporates an energy budget into the optimization procedure for choosing which subset of neurons to split, which it approximately solves by a continuous relaxation, and (ii) avoids doing exact eigendecomposition to extract the minimum eigenvalue (splitting index) but instead replaces it with a more efficient SGD on the Rayleigh quotient.

Evaluation:
--Evaluations are done on variants of MobileNet on CIFAR-100 and ImageNet (the latter would be infeasible without the approximation scheme). The proposed approach appears to get better tradeoff between accuracy and FLOPs in these cases. In practice the non-energy aware "vanilla" networks do tend towards models that are small in size (fewer parameters) but are not necessarily low in energy consumption.
--There is new material here, although I find the novelty a bit limited (e.g. only an additional constraint compared to the original approach of Wu et al and addressing a clear scalability issue with the original work, i.e. eigendecomposition of a matrix, with what seem straightforward approximations, ). The empirical results in Table 1 and 2 seem solid, but I'm not familiar enough with past results in this area  to evaluate their significance.

**Experience Assessment:**

I do not know much about this area.

**Review Assessment: Checking Correctness Of Derivations And Theory:**

I did not assess the derivations or theory.

**Review Assessment: Checking Correctness Of Experiments:**

I assessed the sensibility of the experiments.

**Review Assessment: Thoroughness In Paper Reading:**

I read the paper at least twice and used my best judgement in assessing the paper.

---

> ### Author Response · Authors · 2019-11-15
> **Response to Review #2**
>
> We thank R#2 for the time and comments. Here is our reply to your problems. We hope our work can win stronger support from you to allow us to contribute better to the field. Please let us know if anything can be done to help clarify.
>
> Question: "There is new material here, although I find the novelty a bit limited"
>
> [Since the other reviewers have the same question, we use the same answer for all the reviewers here]
>
> We understand the perception that our work does not feel novel compared to w.r.t. the original splitting descent paper. But splitting descent is a very new and recent work, with the key weakness on scalability to very large models. Our work is a timely contribution to this urgent issue. Also, our algorithm is indeed a very novel approach for the problem of energy-aware architecture optimization (no algorithms of a similar kind exist for this problem; original splitting descent was not energy-aware).  We also believe it is non-trivial to put together our techniques to nicely solve the scalability problem, including our knapsack formulation, Rayleigh gradient, and the ideas of backpropagating on the auxiliary network that allows us to avoid using for loops. We think the application of these techniques in the new content is novel.
>
> More elaboration:
>
> 1) Compared with the original splitting descent paper, our work is *very important*. The original splitting descent *does not work* for large scale settings like ImageNet due to the computation and memory constraint. Our work addresses this key issue nicely and timely. Without our work, splitting descent (and all the potential new algorithms that it may imply) can not work in very large scale settings. We hope the reviewer can kindly consider the fact that methodological novelty is not the only factor to judge the value of a paper (especially when it comes to deep learning). If every work like ours gets rejected, the field can not grow.
>
> 2) Our work is *very novel* to the field of energy-aware architecture optimization, which uses energy metrics to improve DNNs structures. All most existing works are either black-box algorithms (e.g., evolutionary, or reinforcement learning) or using network pruning. Methods that split and grow neurons progressively like our method is *very un-common* in the field of energy-aware deep learning. And we are *the first work* that shows spitting/growing-based methods indeed work empirically for energy-aware NAS on large scale.  We truly want to share this new message with the community and hope it motivates new works and ideas. We would appreciate the reviewers can kindly take this into consideration.
>
> 3) Our key techniques, including the knapsack formulation, the Rayleigh-quotient gradient, and the auto-differentiation trick, are all essential steps for making splitting descent scalable. Yes, these are existing techniques from textbook, but putting them together to solve this problem is new (since there have been no similar works like ours in the literature). We do not believe it is obviously trivial to derive and put all of them in the right place. For example, consider our step of constructing the auxiliary network using auto-diff to calculate the Rayleigh gradient all the neurons simultaneously without using for loop. We think it is indeed a new and essential step that allows us to achieve high scalability (which is at least not obvious to the authors themselves in the beginning).
>
>
> Question: "The empirical results in Table 1 and 2 seem solid, but I'm not familiar enough with past results in this area to evaluate their significance."
>
> Reply: To the best of our knowledge, our results demonstrate SOTA top-1 accuracy on similar flops regime using MobileNets. We believe our results are significant. Since MobileNets are strong baselines, which are specifically hand-designed and heavily tuned to optimize accuracy under a flops-constrain on the ImageNet dataset. Meanwhile, our method also outperforms SOTA pruning methods such as AMC.

---

### Official Review · AnonReviewer4 · 2019-11-11
**Official Blind Review #3**

**Rating:** 3

**Review:**

The paper is well written and flows very well. The idea is straightforward and easy to understand. Here are my feedbacks:

--Method--

Methodology-wise, the novelty is somewhat limited. The main technical contributions are: 1) formulate linear programming to reduce energy costs, and the formulation of linear programming is straightforward. 2) the claimed main contribution is an application of Rayleigh-Quotient gradient descent to approximate the eigenvalue calculations in the model proposed by Wu et al (2019).

I believe computing the eigenvalue shall not be the only way to solve Eq.(3), e.g. using gradient-based method or Lagrange. My main question is whether computing the global optimum really matters? Since your method is still essentially an approximation algorithm, which may break the optimality condition here. From the experimental results, it seems that your approximation is quite close to the analytical solution. Therefore, it is unclear computing a global optimum really matters here.

I tried to buy the idea of escaping the saddle point with splitting (it indeed sounds straightforward and reasonable). It will be great if the author can add some empirical experiments to verify it.

--Content--
The entire section2 is at reviewing prior works, and I believe you should cut down the content here.

--Experiments--
1. Diversity of your tests: the author main uses MobileNet in testing their ideas. It seems that their method starts at MobileNet, then tries to improve it. I suggest authors adding other recent works, e.g. FBNet, MobileNetV3, into their tests to diversify the types of networks.

2. Results variations: all figures, e.g. fig.3, fig.4, in the paper lack plotting their results variations. Since the optimization may converge to different local optimum, it is more persuasive to show their performance variations.

3. The final results are not surprising: in table.1 and table.2, as far as I'm aware, the mainstream accuracy for ImageNet under the mobile setting should be 75% top-1. And the SOTA top-1 accuracy on ImageNet is around 85.5%. Table.1 and table.2 do somewhat show the effectiveness of your method, but its significance is limited especially considering the limited novelty of methodology.

Minors:
In Fig.3 k = 6, it seems vanilla splitting is better than accuracy and parameters, except for 0.2 log higher flops. I don't think this makes a compelling case here.

In Fig.4, from flops 7 ~ 9, your results are similar to Bn especially accuracy > 0.6. When accuracy < 0.6, it is less interesting, and I believe the improvement should be huge.

Fig.5, could you please compare the time using MAGMA from NVIDIA? I had some experiences in implementing the LAPACK and BLAS on GPUs, and it should not be that slow.

Overall, this paper has some interesting results, which shows the eigenvalue can be approximated by Rayleigh-Quotient gradient descent, and show positive improvement. However, the methodological and experimental results can definitely be strengthened. The author may consider proving that achieving the global optimum on Eq.(3) really matters, then motivate the methodology. Thank you.


**Experience Assessment:**

I have published one or two papers in this area.

**Review Assessment: Checking Correctness Of Derivations And Theory:**

I assessed the sensibility of the derivations and theory.

**Review Assessment: Checking Correctness Of Experiments:**

I carefully checked the experiments.

**Review Assessment: Thoroughness In Paper Reading:**

I read the paper at least twice and used my best judgement in assessing the paper.

---

> ### Author Response · Authors · 2019-11-15
> **Response to Review #3 (1)**
>
> Thanks for your time and comments. We do respectfully disagree with some of your understanding and judgment (see below). We hope the reviewer could kindly reconsider your evaluation after reading our response. Please definitely let us know if there is anything we can do to help further clarification.
>
> Question: "Methodology-wise, novelty is somewhat limited"
>
> [Since the other reviewers have the same question, we use the same answer for all the reviewers here]
>
> Reply: We understand the perception that our work does not feel novel compared with w.r.t. the original splitting descent paper. But splitting descent is a very new and recent work, with the key weakness on scalability to very large models. Our work is a timely contribution to this urgent issue. Also, our algorithm is indeed a very novel approach for the problem of energy-aware architecture optimization (no algorithms of a similar kind exist for this problem; original splitting descent was not energy-aware).  We also believe it is non-trivial to put together our techniques to nicely solve the scalability problem, including our knapsack formulation, Rayleigh gradient, and the ideas of backpropagating on the auxiliary network that allows us to avoid using for loops. We think the application of these techniques in the new content is novel.
>
> More elaboration:
>
> 1) Compared with the original splitting descent paper, our work is *very important*. The original splitting descent *does not work* for large scale settings like ImageNet due to the computation and memory constraint. Our work addresses this key issue nicely and timely. Without our work, splitting descent (and all the potential new algorithms that it may imply) can not work in very large scale settings. We hope the reviewer can kindly consider the fact that methodological novelty is not the only factor to judge the value of a paper (especially when it comes to deep learning). If every work like ours gets rejected, the field can not grow.
>
> 2) Our work is *very novel* to the field of energy-aware architecture optimization, which uses energy metrics to improve DNNs structures. All most existing works are either black-box algorithms (e.g., evolutionary, or reinforcement learning) or using network pruning. Methods that split and grow neurons progressively like our method is *very un-common* in the field of energy-aware deep learning. And we are *the first work* that shows spitting/growing-based methods indeed work empirically for energy-aware NAS on large scale.  We truly want to share this new message with the community and hope it motivates new works and ideas. We would appreciate the reviewers can kindly take this into consideration.
>
> 3) Our key techniques, including the knapsack formulation, the Rayleigh-quotient gradient, and the auto-differentiation trick, are all essential steps for making splitting descent scalable. Yes, these are existing techniques from the textbook, but putting them together to solve this problem is new (since there have been no similar works like ours in the literature). We do not believe it is obviously trivial to derive and put all of them in the right place. For example, consider our step of constructing the auxiliary network using auto-diff to calculate the Rayleigh gradient all the neurons simultaneously without using for loop. We think it is indeed a new and essential step that allows us to achieve high scalability (which is at least not obvious to the authors themselves in the beginning).

---

> ### Author Response · Authors · 2019-11-15
> **Response to Review #3 (2)**
>
>  **I believe computing the eigenvalue shall not be the only way to solve Eq.(3), e.g. using gradient-based method or Lagrange;**
>
> Since Eigen-problem is equivalent to Eq (3) in the zero-step size limit, our algorithm is already in some sense doing gradient descent for Eq (3). Brute-forcefully applying gradient descent can be problematic because the gradient of the two off-springs would cancel with each other and yield zero gradients (because they move along opposite directions). As shown in the theory of the original splitting descent paper, the structural descent is a second-order update. Eigen-formulation avoids this problem because it allows us to analytically derive the gradient, which requires to look at some (partial) second-order terms of the objective function in Eq(3).
>
> To put it in another way,  if someone tries to derive a gradient descent algorithm for Eq (3), they will end up with something similar or equivalent to our method, after careful mathematical derivation.
>
> We hope this makes it clear that Eq (3) is not trivial mathematically and theoretical thinking is needed. For example, without deriving the eigen-formulation, we would need to optimize the weights w and size m, which makes the problem complicated.
>
> Issue of "global vs. local optimality"
>
> Following our comments above, we do not think there is an issue of global vs. local optimality here. We are already doing a type of gradient descent on Eq (3), for which the global optimum happens to be the only stable local optimum thanks to its equivalent to Eigen-problem.
>
> Question: "I tried to buy the idea of escaping the saddle point with splitting (it indeed sounds straightforward and reasonable). It will be great if the author can add some empirical experiments to verify it."
>
> Reply: We believe this question has already been fully answered in the original splitting descent paper. For example, Figure 1 in that paper offers very intuitively and convincing illustrations that might alleviate R#3?s concerns.
> Our main focus is on scaling the algorithm to large-scale settings. We only cite and explain their point on this aspect.
>
>
> Question: "adding other recent works, e.g. FBNet, MobileNetV3, into their tests to diversify the types of networks."
>
> Reply:  We choose MobileNetV1 and MobileNetV2 as our testbed mainly due to their popularity and nice accuracy-flops tradeoff. And MobileNetV1 and MobileNetV2 together define all basic blocks/layers of many recent works such as FBNet and MobileNetV3. We expect that the results will be similar on FBNet and MobileNetV3.
>
> Question:  "it is more persuasive to show their performance variations."
>
> Reply:  We provide ablation studies in section 5.4 (see Figure 6).
>
> Question:  "The final results are not surprising: in table.1 and table.2, as far as I'm aware, the mainstream accuracy for ImageNet under the mobile setting should be 75% top-1. And the SOTA top-1 accuracy on ImageNet is around 85.5%."
>
> Reply: The main goal of this paper is to provide a way to grow networks that work in practice. Therefore, our method should be compared with existing methods using the same meta-architectures (e.g. same depth, kernel size, resolutions).  To the best of our knowledge, our results are the current SOTA top-1 accuracy on ImageNet using *MobileNetV1* and *MobileNetV2* with similar Flops compared to expert-designed MobileNets and pruning methods.
>
> Question:  "In Fig.3 k = 6, it seems vanilla splitting is better than accuracy and parameters, except for 0.2 log higher flops. I don't think this makes a compelling case here."
>
> In this case, all networks are 1) not very deep; 2) trained on a small CIFAR10 dataset. That said, the differences shouldn't be super significant. However, the results are shown in Figure 3 clearly demonstrate that our method achieves better accuracy-flops trade-off than the vanilla splitting approach.
>
> Question: "In Fig.4, from flops 7 ~ 9, your results are similar to Bn especially accuracy > 0.6. When accuracy < 0.6, it is less interesting, and I believe the improvement should be huge."
>
> Reply: We believe it?s already significant enough to show that our method outperforms pruning methods. Pruning methods have already taken advantage of using knowledge from a pre-trained over-parameterized model, while our method training small models from scratch.
>
> Question: "Fig.5, could you please compare the time using MAGMA from NVIDIA? I had some experiences in implementing the LAPACK and BLAS on GPUs, and it should not be that slow."
>
> Reply:  Thanks for your suggestions. We note that the key bottleneck of the vanilla splitting approach is the requirement of calculating all splitting matrices, which causes a space complexity O(nd^2) (n: number of neurons, d: dimension of each neuron).  This makes it *impossible* to implement the algorithm on GPUs for modern neural networks with thousands of neurons, mainly due to the explosion of GPU memory.

---

### Decision · Program_Chairs · 2019-12-19

**Decision:**

Reject

**Comment:**

This paper extends previous work on searching for good neural architectures by iteratively growing a network, including energy-aware metrics during the process. There was discussion about the extent of the novelty of this work and how well it was evaluated, and in the end the reviewers felt it was not quite ready for publicaiton.